# Polyoxometalate-Decorated Gold Nanoparticles Inhibit β-Amyloid Aggregation and Cross the Blood–Brain Barrier in a µphysiological Model

**DOI:** 10.3390/nano13192697

**Published:** 2023-10-03

**Authors:** Marta Perxés Perich, Sujey Palma-Florez, Clara Solé, Sara Goberna-Ferrón, Josep Samitier, Pedro Gómez-Romero, Mònica Mir, Anna Lagunas

**Affiliations:** 1Catalan Institute of Nanoscience and Nanotechnology(ICN2) CSIC and BIST, Campus UAB, Bellaterra, 08193 Barcelona, Spain; 2Nanobioengineering Group, Institute for Bioengineering of Catalonia (IBEC), Barcelona Institute of Science and Technology (BIST), 08028 Barcelona, Spain; 3Materials Chemistry and Catalysis, Debye Institute for Nanomaterials Science, Utrecht University, Universiteitsweg 99, 3584 CG Utrecht, The Netherlands; 4Instituto Universitario de Tecnología Química (CSIC-UPV), Universitat Politècnica de València, Avda. De los Naranjos s/n, 46022 Valencia, Spain; 5Biomedical Research Networking Center in Bioengineering, Biomaterials, and Nanomedicine (CIBER-BBN) Monforte de Lemos 3-5, Pabellón 11, 28029 Madrid, Spain; 6Department of Electronics and Biomedical Engineering, University of Barcelona, Martí i Franquès 1, 08028 Barcelona, Spain

**Keywords:** nanovehicle, gold nanoparticles, polyoxometalates, β-amyloid, blood–brain barrier organ-on-a-chip

## Abstract

Alzheimer’s disease is characterized by a combination of several neuropathological hallmarks, such as extracellular aggregates of beta amyloid (Aβ). Numerous alternatives have been studied for inhibiting Aβ aggregation but, at this time, there are no effective treatments available. Here, we developed the tri-component nanohybrid system AuNPs@POM@PEG based on gold nanoparticles (AuNPs) covered with polyoxometalates (POMs) and polyethylene glycol (PEG). In this work, AuNPs@POM@PEG demonstrated the inhibition of the formation of amyloid fibrils, showing a 75% decrease in Aβ aggregation in vitro. As it is a potential candidate for the treatment of Alzheimer’s disease, we evaluated the cytotoxicity of AuNPs@POM@PEG and its ability to cross the blood–brain barrier (BBB). We achieved a stable nanosystem that is non-cytotoxic below 2.5 nM to human neurovascular cells. The brain permeability of AuNPs@POM@PEG was analyzed in an in vitro microphysiological model of the BBB (BBB-on-a-chip), containing 3D human neurovascular cell co-cultures and microfluidics. The results show that AuNPs@POM@PEG was able to cross the brain endothelial barrier in the chip and demonstrated that POM does not affect the barrier integrity, giving the green light to further studies into this system as a nanotherapeutic.

## 1. Introduction

Alzheimer’s disease (AD) is a severe, chronic, and progressive neurodegenerative disorder associated with memory loss, impairment cognition, and behavioral alterations [1]. AD presents neuronal death associated with a combination of neuropathological hallmarks, such as abnormalities in amyloid processing, which usually implies the extracellular accumulation of β-amyloid (Aβ) plaques or fibrils [2,3]. These fibrils are derived from Aβ monomers, which can associate and form aggregates, resulting in toxic elements for axons and dendrites [4]. Aβ aggregation has been thoroughly studied, and numerous ligands have been synthesized to target it and inhibit its aggregation, as a therapeutic strategy [5,6].

During recent years, countless possible drugs have shown a low effectiveness in the middle and late stages of clinical trials due to their adversity to crossing the blood–brain barrier (BBB), which strictly protects the brain from any blood-circulating molecules that intend to enter it [7]. All these failures result in a high cost for pharmaceutical companies that causes them to abandon their drug development studies against AD [8]. Currently, Aβ-directed antibodies are maybe the most successful AD therapy. Examples such as aducanumab have been prominent in recent years; however, the commercialization of the former was rejected by several drug agencies, while Lecanemab showed highly significant results in phase 3 but attracted controversial opinions on its previous performance [9,10,11]. The latter has recently been approved by the FDA, and it is the first amyloid-β-targeting drug to be approved [12].

Nanovehicles are attractive for drug delivery because they can be beneficial for drug stability and solubility, as well as helping the penetrate biological barriers as the BBB, allowing drugs to reach their necessary effective concentration in the brain [13], which, as mentioned before, is one of the main limiting factors for drug efficiency in the case of AD. A commonly studied nanovehicle for drug delivery is the gold nanoparticle (AuNP), as these particles are biocompatible and their surface is highly tunable, allowing further functionalization with bioactive ligands [14,15,16,17,18,19]. For example, functionalization with polyethylene glycol (PEG) confers stability in biological media and reduces protein absorption, increasing the residence time in the blood [20,21,22].

Here, we present a tri-component nanohybrid system, based on gold nanoparticles (AuNPs) functionalized with PEGs and with polyoxometalates (POMs), AuNPs@POM@PEG. Polyoxometalates (POMs) are early transition metal oxide anionic clusters (mainly Mo, W, and V) with a well-defined 0-D molecular structure, which can vary in composition, structure, and size [23,24]. They are very well known for their redox activity and are widely used in the field of catalysis and energy storage [25]. Lately, POMs are also attracting attention in the biomedical field, as they can disrupt tumor growth while not being cytotoxic in non-tumoral cells, and can act as antiviral agents [26,27]. In AD, POMs are potentially attractive as their size and negative charge allow them to bind to the Aβ monomer, lowering the concentration of the free monomer and shifting the equilibrium away from Aβ fibrillization [28,29]. Further studies demonstrated that the inclusion of functional groups as histidine-chelating metals or chiral compounds in the structure of POMs significantly increased the inhibition of Aβ aggregation [30,31]. Here, we used the β_2_ isomer of the monolacunary Keggin ([β_2_-SiW_11_O_39_]^8−^; abbreviated POM), as it had been previously used in Aβ aggregation inhibition assays [28,29].

Despite the great advantages that AuNPs combined with PEG and POMs could provide as therapeutic agents for AD treatment, there are only a few examples in the literature of AuNPs being used in combination with POMs for the treatment of AD. Gao et al. constructed AuNPs@POMD-pep, which used a POM with a Wells–Dawson structure and a β-sheet breaker peptide (N-Ac-CLPFFD) that were able to reduce to the half the aggregation of Aβ, when used at 40 nM concentration [32]. In another case, POMs were used as linkers between Au nanorods (AuNRs) and the Aβ-targeted peptide inhibitor Aβ_15–20_ [33]. POMs have also been combined with CeO NPs, showing Aβ aggregation inhibition as well as Aβ degradation and reduced ROS levels [34].

Moreover, the evaluation of POMs, either alone or combined with nanosystems, as potential candidates for AD treatment includes the use of animal models, which, as well as raising ethical concerns, leads to a lack of predictability due to the physiological differences between humans and animals. Indeed, the animal models used for AD are almost exclusively limited to transgenic mice that, while capable of generating amyloid plaques, do not develop AD, thus making it difficult to test the efficacy of the drugs evaluated in these models [35].

In the field of drug discovery, it is becoming necessary to look for inexpensive, quick, and animal-free alternatives for the early provision of these possible AD therapeutic agents. Organ-on-a-chip (OoC) systems are an emergent technology resembling physiological and pathological conditions on a micrometric scale. OoC offers the possibility to include different types of cells as well as an extracellular matrix, cell–cell interactions, and dynamic conditions [36]. Previously, OoC has been used for mimicking the BBB for multiple applications, such as brain tumour research, the identification of neurotoxic compounds, and drug screening, among others [37]. Our group has previously developed a BBB-oC system capable of monitoring the drug permeability in real time through the fluorescence labelling of AuNPs [38].

Here, we combine the excellent advantages offered by AuNPs with PEGs and the POM, to achieve an efficient nanohybrid system for the inhibition of Aβ aggregation for AD treatment. We synthetized and characterized a AuNPs@POM@PEG nanosystem and evaluated its performance as an inhibitor of Aβ aggregation via thioflavin T (ThT) assays. To assess the biocompatibility and targeting feasibility of AuNPs@POM@PEG as an AD therapeutic agent, its cytotoxicity and its permeability through the BBB were tested in an in vitro BBB-oC model containing a 3D human neurovascular cell co-culture.

## 2. Results and Discussion

### 2.1. Synthesis and Characterization of AuNPs@POM@PEG

Citrate-capped spherical gold nanoparticles (AuNPs@SC) were synthesized following the Turkevich method, which is described in detail in the experimental section [39,40,41]. In this reaction, a gold precursor (HAuCl_4_) is mixed with sodium citrate (SC) at a high temperature. SC acts as both a reducing agent and a surfactant, allowing tunable particle sizes to be obtained by changing the ratio between SC and HAuCl_4_ [40,42]. We obtained a AuNPs@SC system of 16.0 ± 0.8 nm in diameter, as determined via electron microscopy (Appendix A). 

Our polyoxometalate choice was the β_2_ isomer of the monolacunary Keggin β_2_-K_8_SiW_11_O_39_·12H_2_O (POM) (Appendix A [43]), as it has previously been used in Aβ aggregation inhibition assays and demonstrated a good performance [28]. POM was synthesized from sodium tungstate and sodium silicate using the procedure described by Tézé et al. [44]. The purity of the POM was assessed via infrared spectroscopy (IR) and via cyclic voltammetry (CV). The β_2_-K_8_SiW_11_O_39_ IR spectrum (Figure 1a, Appendix A) shows characteristic peaks at 989.5 [ʋ s (W–Od)], 943.2 [ʋ as (W–Od)], 872.9 [ʋ as (Si–Oa)], 852.5 [ʋ as (W–Ob–W)], 790.8, and 713.7 [ʋ as (W–Oc–W)] cm^−1^ [45], where Oa, Ob, Oc, and Od represent the central, corner-sharing, edge-sharing, and terminal oxygen atoms of the Keggin structure (Appendix A). Figure 1b shows the CV in 1 M sodium acetate buffer (pH 4.7) of the [SiW_11_O_39_]^8−^ POM and its parent anion [SiW_12_O_40_]^4−^ for comparison. The [SiW_12_O_40_]^4−^ anion is stable in acid solutions with a pH of less than 5 [46]. As reported previously [47], Figure 1b demonstrates that the reduction of [SiW_12_O_40_]^4−^ under a pH < 5 occurs via two initial 1e^−^ steps (processes I and II), followed by an overall 2e^−^ step (process III). On the other hand, the CV of [SiW_11_O_39_]^8−^ at pH 4.7 shows two reduction processes (process I and II) assigned to two overall 2e^−^ redox couples. From Figure 1b, we can observe that (i) the potentials in the CV for the first two redox processes of the [SiW_11_O_39_]^8−^ anion are more negative than those observed for [SiW_12_O_40_]^4−^, and (ii) process I in [SiW_11_O_39_]^8−^ becomes irreversible due to the formation of ion pairs with the cations of the electrolyte (Na^+^) at a high 1 M acetate buffer ionic strength. These observations are consistent with the literature [48].

The POM was attached to the surface of AuNPs through a ligand exchange reaction (Figure 1) promoted by the weak interaction between SC and gold and the excess of POM in the solution [49]. It has been reported that this lacunary POM binds to silver nanoparticles via the pentadentate aperture, with five oxygen atoms directly bound to the surface [50]. Analogously, we assume that a similar chemical bond interaction may occur in the case of AuNPs. The reaction conditions were optimized through the combination and modification of different methods reported in the literature, as described in the Methods section [50,51,52,53]. As depicted in Figure 2a, in contrast to the IR spectra of AuNPs@SC, the IR spectra of AuNPs@POM clearly display the presence of the main IR peaks characteristic of β_2_-K_8_SiW_11_O_39_· (Figure 1a), demonstrating the correct incorporation of POM on the AuNP (see also Appendix A). Then, AuNPs@POM was functionalized with two types of thiolated PEGs, O-methylated and O-carboxylated, through the bonding of the thiol groups to gold. The thiol O-methylated PEG (SH-PEG-OMe) was used to provide colloidal stability by introducing neutral charges, and the thiol carboxylated PEG (SH-PEG-COOH) allowed us to further functionalize the nanosystem. We characterized the resulting AuNPs@POM@PEG via infrared (IR) spectroscopy. The IR spectra of AuNPs@POM@PEG (Figure 2a and Appendix A) not only display the characteristics peaks from POM, but also additional peaks at 1350.2, 1298.1, and 1095.6 cm^−1^ due to PEG vibrations (C-H bending and O–H and C–O–H stretching, respectively [54]), demonstrating that PEG functionalization was achieved while preserving POMs. In addition, Figure 2b shows an electron microscopy image of AuNPs@POM@PEG exhibiting a spherical shape and a size of 17.7 ± 2.3 nm (Appendix A).

Ultraviolet–visible spectroscopy (UV–Vis) is a very widely used technique for characterizing spherical gold nanoparticles thanks to the surface plasmon resonance (SPR) peak at around 520 nm. It can be used to quickly determine the concentration of AuNPs, and it is also a good indicator of any changes occurring at the AuNP surface [55,56]. Figure 2c shows a red shift in the SPR signal of the AuNPs after ligand exchange, caused by the increase in the refraction index after POM and PEG binding on the AuNP surface [50]. 

As mentioned, PEG functionalization brings stability to the nanosystem in solutions that present an elevated ionic/salt content, such as physiological media [57,58]. Figure 2d demonstrates that AuNPs@POM (without PEG) rapidly aggregates in PBS, showing a weak SPR peak in UV–Vis spectra. After PEG functionalization, the AuNPs@POM@PEG SPR peak remains intact, only slightly red-shifted due to the increase in the dielectric constant of PBS [59]. The UV–Vis spectra did not suffer any major changes after 14 weeks (Appendix A), demonstrating the good stability of AuNPs@POM@PEG in PBS.

X-ray photoelectron spectroscopy (XPS) was used to confirm the presence of POM in the AuNPs@POM@PEG and to determine the oxidation state of the gold (Appendix A). The Au 4f core-level spectrum resolves into two spin-orbit components. The Au 4f7/2 and 4f5/2 peaks occur at a BE of 84.05 and 87.75 eV, respectively. These values indicate that gold is present only in the metallic form. The W4f spectrum shows a W4f 5/2 and W4f 7/2 doublet with the binding energies of 36.30 and 38.47 eV, respectively, which correspond to W(VI) species present in POM [60]. XPS was also used to analyze the POM coverage of the AuNPs@POM@PEG, as it gives quantitative information about the composition of the surface when using different ratios of POM/AuNPs and PEG/AuNPs, obtaining a W coverage of around 5.5%.

In addition, to prove the incorporation of different ligands into the AuNP surface, the hydrodynamic diameter (Dh) and zeta potential (pZ) values were determined. As shown in Table 1, the Dh changed from 33 nm to 60 nm due to the PEG and POM functionalization on the AuNPs@SC surface. Further, Table 1 shows pZ changes in the charge of the nanoparticles, where AuNPs@SC displays more negative charge after POM conjugation to form AuNPs@POM, due to the high negative charge (−8) of POM. On the other hand, the addition of SH-PEG-OMe and SH-PEG-COOH to AuNPs@SC maintains the charge, as SH-PEG-COOH has also a negative charge in solution, and SH-PEG-OMe has no formal charge. Finally, AuNPs@PEG@POM reveals a Dh increase and pZ changes towards more negative values after POM addition. In conclusion, changes in the hydrodynamic diameter (Dh) and zeta potential (pZ) values confirmed the functionalization of the AuNPs@SC surface with PEG and POM. 

### 2.2. Evaluation of the Inhibition of Aβ Fibrillization In Vitro by AuNPs@POM@PEG

The Aβ aggregation was characterized and monitored using the Thioflavin T (ThT) fluorescence assay. This technique is based on the binding of ThT to the β-sheet conformation of aggregated Aβ, giving a fluorescence emission at 480 nm that is proportional to the number of fibrils formed [61,62]. To evaluate the inhibition of Aβ aggregation in vitro, we incubated Aβ_1–42_ peptide with different concentrations of AuNPs@POM@PEG (1, 2.5 and 5 nM), with the AuNPs@PEG control at 4 nM, and with free-standing POM at 100 µM, as used in previously reported assays [28], for 2 h at 37 °C. In addition, controls with ThT with glycine and glycine buffer only were also conducted.

Figure 3a shows the kinetics of Aβ aggregation for each condition over 2 h at 37 °C. Specifically, in untreated Aβ peptide, the fluorescence intensity increases, which can be attributed to the growth of Aβ fibers or to an increase in the total number of fibres that finally reaches a plateau, as described for the classic amyloidogenic process [63]. In contrast, samples treated with AuNPs@POM@PEG did not display this fluorescence increase, which is attributed to their inhibitory action on Aβ aggregation. We observed that the different concentrations of AuNPs@POM@PEG used did not show significant differences in their degree of inhibition, and that both POM and AuNPs@PEG alone show anti-aggregation activity, as previously described [18,28,30], although at lower levels than when combined in AuNPs@POM@PEG. The results demonstrated how the anti-aggregation potential of POM improves over 50% through its conjugation with AuNPs. It should be noted that AuNPs@POM (without PEG) could not be included in the study as they are not stable enough in the high-salinity medium used in the assay. However, previous studies reported that PEGs can also bind to the Aβ protein and stabilize its native structure, therefore suppressing fibrillization in its initial stage [64]. This may suggest a synergetic behavior of the three components of the nanosystem (AuNPs, POM, and PEG), enhancing its performance. Aβ treated with POM and AuNPs@PEG exhibits an ascendent kinetic behavior in the early stages of the Aβ aggregation process, while samples with AuNPs@POM@PEG show a negative tendency at the beginning, as has been previously reported [18]. Finally, controls of ThT with glycine and glycine buffer present the lowest fluorescence signal, proving that there is no interference in the fluorescence measurements. Figure 3b shows the aggregation percentage of each condition relative to the untreated Aβ. We observe a strong inhibitory effect of AuNPs@POM@PEG with an inhibition of about 75% in fibril formation after treatment compared with the untreated Aβ sample, surpassing the performance of previously reported AuNPs-POMs including the specific Aβ-targeted peptide inhibitor Aβ_15–20_ [30,36].

### 2.3. Evaluation of the Cytotoxicity and Permeability of AuNPs@POM@PEG across the Blood–Brain-Barrier-on-a-Chip (BBB-oC) In Vitro Model

As a possible candidate for AD treatment, AuNPs@POM@PEG should be able to cross the BBB. Therefore, we evaluated the permeability of AuNPs@POM@PEG in an animal-free, microphysiological model of the human BBB. Our BBB-oC system consists of a 3D scaffold made of fibrin hydrogel in which human astrocytes and pericytes are embedded (brain chamber), and where they are in direct contact with human endothelial cells in a lateral channel (blood channel), which is separated from the main chamber through microposts (Figure 4a) [38]. Before testing the permeability of the nanosystem, we conducted cytotoxicity assays on the three different cell types used in the BBB-oC model to make sure that the administered dose of nanoparticles was not harmful to the cells. Using two complementary methods, live/dead assay and MTS, we screened the cell viability in the presence of four different AuNPs@POM@PEG concentrations ranging from 0.5 to 5 nM based on the previous reported cytotoxicity limit of AuNPs@POM [26].

After 24 h of incubation, the AuNPs@POM@PEG are not cytotoxic below 2.5 nM. The defined ISO 10993-5 standard dictates that a material is not cytotoxic if the cell viability is above 70 % after at least 24 h of incubation [26]. The percentages of cell viability calculated from both the MTS and live/dead assay are shown in Figure 5. As expected, both methods show the same trend: a concentration-dependent behavior, where a higher concentration decreases the viability of the cells after treatment. For both human pericytes and astrocytes, the cell viability is above 75% for all concentrations except 5 nM AuNPs@POM@PEG, in which the viability is still higher than 50%, indicating a mild cytotoxicity. Human endothelial cells were more affected by AuNPs@POM@PEG, and showed a mild cytotoxicity between 5 and 2.5 nM, but they were not cytotoxic with 1 and 0,5 nM of concentration, with which they presented over 80% of viability. Appendix A shows some live/dead images of the different treatments of endothelial cells where differences in the living-cell density can be observed.

To evaluate the BBB permeability of the AuNPs@POM@PEG, and according to the cytotoxicity results, 2.5 nM AuNPs@POM@PEG were fluorescently labelled with Alexa fluor 647 and injected into the blood channel of the chip. Fluorescent images were recorded for 1 h to determine the permeability coefficient (P). Also, fluorescently labelled 2.5 nM AuNPs@PEG were used to determine whether POM molecules had any effects on the BBB crossing. Figure 4b reveals that both nanoparticles were effectively internalized towards the barrier. The permeability values calculated were found within the same range as those previously reported for nanoparticles with similar features in terms of material and size in analogous BBB-oC devices [38]. The *p* values of AuNPs@POM@PEG (3.47·10^−6^ cm/s) and AuNPs@PEG (3.07·10^−6^ cm/s) showed no significant differences, thus indicating that POM had no effects on the barrier integrity. Finally, the hydrogel of the brain channel was dissolved with TRIzol^®^, used for its capacity for sample homogenization, and then observed via STEM. AuNPs@POM@PEG were found in the dissolved hydrogel (Figure 4c), confirming that the nanosystem crossed the endothelial barrier in the BBB-oC; additionally, EDX showed the presence of gold in the sample and, therefore, the presence of AuNPs (Figure 4d).

Several studies have demonstrated the efficient transport of AuNPs across the BBB [65], showing that the cellular uptake of AuNPs depends on multiple factors, such as the size, surface charge, concentration, and type of coating, among others [66,67,68,69]. For example, 5 nm AuNPs are internalized via passive diffusion through the formation of transient pores in the cell, without altering the cell membrane. AuNPs with sizes from around 20 to 50 nm are preferably internalized via endocytic-like processes, while those greater than 70 nm are not efficiently internalized [67,69,70]. As the cell membrane has a negative charge, positively charged nanoparticles are expected to show a higher cellular uptake than negatively or neutrally charged nanoparticles. However, in vivo studies that evaluated the accumulation in organs according to different sizes (from 1.4 to 200 nm) and charges found that AuNPs of around 20 nm reached the maximum accumulation in the central nervous system, and that negatively charged nanoparticles showed better penetration than positively charged ones [71]. In addition, cationic AuNPs are associated with a higher cellular toxicity, as they can generate transient pores in the cell membrane. It has also been observed that AuNPs can induce hemolysis and platelet aggregation processes [72,73,74]. Therefore, our negatively charged ~18 nm AuNPs@POM@PEG have suitable characteristics, in terms of size and charge, for a good cell internalization, as demonstrated in the BBB-oC assays.

## 3. Conclusions

We produced the tri-component nanohybrid system AuNPs@POM@PEG, based on gold nanoparticles (AuNPs) covered with polyoxometalates (POMs), and we fully characterized it by deploying a set of techniques, such as IR, CV, XPS, STEM, UV–Vis, Dh, and zeta potential. The presence of PEG increased the stability of the nanosystem, which was preserved for 14 weeks, demonstrating a good long-term stability in physiological media. We proved that AuNPs@POM@PEG effectively inhibits β-amyloid aggregation, observing a synergetic effect between AuNPs and POM components that allowed us to reach inhibition levels of up to 75%. The permeability in terms of crossing the BBB of AuNPs@POM@PEG has been studied in a human BBB-oC model. In our cytotoxicity study, we found that AuNPs@POM@PEG could be used at concentrations of up to 2.5 nM, considering human neurovascular cell viability. The results show that AuNPs@POM@PEG can cross the BBB at similar levels to other nanovehicles reported for drug delivery into the brain, and we verified that POM has no impact on the permeability of the nanosystem. Although studies into biodistribution into the brain and assessments of the therapeutic effects of AuNPs@POM@PEG are required, these preliminary results pose AuNPs@POM@PEG as a promising candidate for use in AD treatment via systemic administration, which could be further extended to other NDDs. Therefore, we are confident that this work can inspire future research on the use of POM-functionalized AuNPs for the treatment of NDDs.

## 4. Materials and Methods

### 4.1. Chemicals

All chemicals were purchased from Sigma-Aldrich unless indicated otherwise. HS-PEG-OMe and HS-PEG-COOH MW 5000 Da were provided by JenKem Technology (Plano, TX, USA). Human hippocampal astrocytes, human brain-vascular pericytes, poly-L-lysine, trypsin/EDTA 0.05%, and astrocyte and pericyte media were supplied by Sciencell (Carlsbad, CA, USA). The human brain endothelial cell line (hCMEC/D3) and EndoGRO™ MV cell medium kit were purchased from Merck-Millipore (Billerica, MA, USA), and the MTS reagent from PROMEGA. Aβ Amyloids 1–42 were purchased from Genecust. Alexa 647 hydrazide (1 mg) was obtained from Thermo Fischer Scientific (Waltham, MA, USA). The SYLGARD™ 184 Silicone Elastomer Kit for polydimethyl siloxane (PDMS) was purchased from Ellsworth (Pierce County, WI, USA). Deionized Milli-Q water of 18 MW x cm was obtained using Milli-Q^®^ purification equipment from Merck Millipore (Darmstadt, Germany), using previously deionized conventional water (acid and basic columns).

### 4.2. POM Synthesis

The potassium salt of the β2 isomer of the monolacunary 11-tungstosilicate β_2_-K_8_SiW_11_O_39_·(POM) was synthesized at a quarter scale, as previously described [44]. In brief, a sodium tungstate solution (45.5 g of Na_2_WO_4_·2H_2_O in 75 mL H_2_O) was cooled to approximately 5 °C, and 41.25 mL of 4M HCl was added gradually in 1 mL portions over 20 min, to reach a pH of 7.8. The solution was removed from the ice bath, and a solution of 2.5 g of silicate (Na_2_SiO_3_·5H_2_O) in 25 mL of H2O was poured into the sodium tungstate solution. The pH was adjusted to 5.5 and maintained for an hour using 4 M HCl. The potassium salt (β_2_-K_8_SiW_11_O_39_, the POM) contaminated with paratungstate B ([H_2_W_12_O_42_]^10−^) was precipitated by the slow addition of 20 g of solid KCl with gentle stirring for 15 min until precipitation was complete and the solid was vacuum filtered. Purification was achieved through dissolving the product in 212.5 mL of water. The insoluble part (the paratungstate contamination) was removed via gravity filtration, and the soluble part (the SiW_11_O_39_^8−^) was precipitated again from the filtrate via the addition of solid KCl (20 g). The resulting precipitate (K_8_SiW_11_O_39_, the POM) was vacuum-filtered, washed with diluted 2M KCl solution (2 portions of 15 mL), and air-dried.

### 4.3. POM Characterization

Infrared spectroscopy was performed using a Bruker Tensor 27 spectrometer (Bruker, Billerica, MA, USA). Spectra were collected at a resolution of 4 cm^−1^. Cyclic voltammetry (CV) was performed in a three-electrode configuration using a Biologic VMP3 multi-channel potentiostat. The glassy carbon working electrode, Ag|AgCl reference electrode, and Pt counter electrode were submerged in a 5 mM POM solution in buffer acetate at pH 4.7. A scan rate of 50 mV·s^−1^ was used.

### 4.4. Gold Nanoparticle Synthesis

All glassware was washed with aqua regia, then rinsed thoroughly with Milli-Q water. The 16 nm gold spheres were synthesized via the sodium citrate reduction method described by Turkevich [40,41]. In brief, 6 mL of HAuCl4 10 mM were added to 112 mL of boiling water in a three-neck round-bottom flask under reflux in an oil bath at 150 °C under vigorous stirring. After the temperature was allowed to stabilize for 10 min, 2 mL of a 60 mM freshly prepared sodium citrate solution was added into the vortex with a syringe. The color immediately shifted from light yellow to transparent, then to dark purple and, finally, to bright red. Five minutes after the citrate solution was added, the solution was taken out the oil bath and allowed to cool down under reflux. The final concentrations were 1 mM sodium citrate and 0.5 mM HAuCl_4_. The as-synthesized gold nanoparticles had a concentration of 1.75 nM and were stored in the dark at room temperature.

### 4.5. AuNPs@POM@PEG Functionalization

In a standard ligand exchange reaction, 120 mg of POM in 2 mL H_2_O was added dropwise to a 40 mL solution of as-synthesized gold nanoparticles under stirring. After 1 h of reaction, 30 µL of PEG-COOH and 10 µL of PEG-OMe (1 mg/200 µL) were added. After 15 min of stirring, the AuNPs@POM@PEG was washed twice via centrifugation (40 min at 8000 rcf). Controls of AuNPs@PEG (without POM) and AuNPs@POM (without PEG) were performed via skipping the addition of POM or PEGs, respectively.

### 4.6. Alexa 647 Functionalization

The AuNPs@POM@PEG was centrifuged, and the pellet was resuspended in 500 µL of buffer MES 0.1 M at pH 5.5 with 1 mg 1-Ethyl-3-(3-dimethylaminopropyl) carbodiimide (EDC) and 2.5 mg N-Hydroxysuccinimide (NHS) and incubated for 15 min under sonication. The excess EDC/NHS was removed via centrifugation (40 min at 8000× *g*). The pellet was resuspended in 500 µL H_2_O and then 10 µL of Alexa 647 hydrazide 1 mg/mL was added, followed by incubation overnight. The following day, the AuNPs@POM@PEG@Alexa647 was washed three times via centrifugation (40 min at 8000× *g*).

### 4.7. Characterization of AuNPs@PEG@POM

UV–Vis spectroscopy was carried out with a Cary 4000 spectrometer (r (Agilent Technologies, Santa Clara, CA, USA) in 1 cm optical path disposable PMMA cuvettes. Attenuated total reflectance Fourier transform infrared spectroscopy (ATR-FTIR) was collected using a Bruker Tensor 27 spectrometer (Bruker, Billerica, MA, USA), 32 running scans were collected at a resolution of 4 cm−1 In the case of AuNPs@POM, the NP suspension was centrifuged and redispersed in a small EtOH volume. The pellet was introduced to the measuring slot, and time was given for the ethanol to evaporate. The UV–Vis and FTIR data were analyzed with Spectragryph software. Dynamic light scattering (DLS) was used to determine the hydrodynamic diameter (Dh) and polydispersity index (Pdi), and the surface charge (zeta potential) was measured via laser Doppler micro-electrophoresis; both used ZetaSizer equipment (Nano ZS, Malvern Panalytical Ltd, Malvern, UK)) with folded capillary cells in water. Electron microscopy was performed with a FEI Magellan 400L XHR SEM (FEI Company, Hillsboro, OR, USA), on high-resolution STEM mode, with a voltage of 20 kV. The sample preparation consisted of the drop deposition of the solutions on a holey carbon mesh grid. XPS measurements were carried out using a SPECS PHOIBOS 150 hemispherical analyzer (SPECS Surface Nano Analysis GmbH, Berlin, Germany) with Al Kα radiation (1486.74 eV), and data were analyzed using CasaXPS software v2.3.16Dev52.

### 4.8. βamyloid Aggregation Inhibition

ThT fluorescence assay. A Corning 384-well black flat-bottom microplate was used. Each well contained 20 µL of glycine buffer 0,1 M pH 8,4, 5 µL of treatment (AuNPs@POM@PEG) in PBS, 20 uL Aβ monomer, and 5 µL ThT 100 µM. Before the experiment, ThT 100 µM was prepared and filtered through 0.22 um. The glycine and the treatment were first added into the wells. Then, previously prepared Aβ aliquots (0.1 mg) were thawed and diluted with HFIP and PBS 1×, to obtain Aβ_1–42_ 44.3 µM. Then, the Aβ solution was added in the wells. Just before measurement, the ThT solution was added, reaching a final volume of 50 µL in each well. The data were read every 5 min using a microplate reader (Infinite M200 PRO). After two hours of incubation at 37 °C, untreated Aβ samples was established as 100%, and the rest of the percentages were calculated with respect to this value. In this way, we normalized the variability in the ThT assay, to be able to combine the results. These experiments were performed N = 5.

### 4.9. Cell Culture

Human endothelial cells (hCMEC/D3) were cultured in T75 culture flasks coated with collagen type I from rat tails (1:20 in PBS buffer) in endothelial medium-supplemented ENDOGRO (Merck) with FGF-2. Human pericytes and astrocytes cells were cultured in T75 culture flasks coated with poly-L-lysine (2μg/cm2 in sterile water) in corresponding pericyte and astrocyte media. All cells were maintained in a humidified incubator at 37 °C and 5% pCO_2_, and the media were changed every two days. Cells were detached using 0.25% trypsin/EDTA for endothelial cells and 0.05% trypsin/EDTA for pericytes and astrocytes.

### 4.10. Cytotoxicity Assays

For the potential application of AuNPs@POM@PEG for NDD, the cytotoxic effect of AuNPs@POM@PEG in human endothelial, astrocyte, and pericyte cells was evaluated. Two complementary assays were performed: MTS and live/dead. The particles had previously been sterilized with UV radiation, and then they were incubated with each of the neurovascular cells for 24 h at 37 °C and 5% pCO_2_. We performed two complimentary assays. On the one hand, the colorimetric assay [3-(4,5-dimethylthiazol-2-yl)-5-(3-carboxymethoxyphenyl)-2-(4-sulfophenyl)-2H-tetrazolium] (MTS) gives information on cell metabolism, based on the reduction of the MTS reagent to a colored dye, carried out by NAD(P)H-dependent dehydrogenase enzymes in metabolically active cells [75]. On the other hand, the live/dead assay consists of staining living cells with calcein AM and dead cells with ethidium homodimer-1 [76]. We also stained cell nuclei with Hoetch, and then cells were observed under a fluorescence microscope, and living cells were counted with ImageJ. Cells were detached, and 12,000 cells were seeded in a coated 96-well plate with 100 µL of the corresponding media for MTS, and 120,000 cells were seeded in a coated 24-well plate with 1 mL of corresponding media for live/dead. Cells were maintained at 37% and 5% pCO_2_. After 24 h, different concentrations of AuNPs@POM@PEG were added to the cells and left to incubate for 24 h under the same conditions regarding temperature and pCO_2_. In addition, live and dead controls were performed with supplemented medium and staurosporine 10 µM, respectively. After 24 h, the supernatant was removed, and the MTS reagent was added to the corresponding cell media. Cells were incubated for 2 h, and the absorbance difference (Abs 490 nm–Abs 655 nm) was recorded using the microplate reader (Infinite M200 PRO). For the live/dead, the supernatant was removed, and 0.5 mL of live/dead solution consisting of 2 µM calcein AM (fluorophore for living cells) and 4 µM EthD-1 (fluorophore for dead cells) and Hoetch 1:1000 (which stains nuclei) were added to the PBS. After 30 min of incubation, cells were washed with PBS and observed using a fluorescent microscope (Olympus IX71), with 10 pictures taken per condition. All experiments were repeated at least three times, with MTS assays having five replicates per condition, and live and dead experiments having two replicates per condition.

### 4.11. BBB-on-a-Chip Device Fabrication

Master molds were obtained from 100 mm silicon wafers using standard photolithography techniques in a clean room environment. In brief, the wafers were activated in the plasma chamber at high mode (10.5 W) for 1 min, and two layers of SU8-3050 photoresist were spun to a total height of 120 μm. After that, the photoresist was soft baked for 25 min and then exposed using an I-line mask aligner to transfer the design in the acetate photomask (JD Photo Data, Hitchin, UK). Finally, the wafers were post-baked, developed for around 20 min and then hard-baked for 30 min at 95 °C and 10 min at 65 °C. Finally, the wafers were silanized for 1 h with trifluorosilane. The PDMS replicas were obtained through mixing a PDMS ratio of 10:1 (base:crosslinking agent), degassed for 45 min, and cured for 2 h at 65 °C. Finally, the single device was bonded with glass coverslips via plasma activation treatment for 30 s at high mode (10.5 W).

### 4.12. BBB-oC Cell Seeding

To obtain a BBB-oC with a neurovascular network, we relied on a procedure described by Campisi and collaborators [77]. On day 0, pericyte and astrocyte cells were trypsinized, and 4 × 10^4^ of each cell line was mixed in a fibrin hydrogel consisting of fibrinogen 3mg/mL in PBS and 100U thrombin in PBS. The mix was injected into the brain channel, and the chip was incubated for 15 min at 37 °C 5% pCO_2_ to allow the hydrogel to polymerize. After polymerization, a mixture of astrocytes and endothelial medium (1:1 ratio) was added into the lateral channels to supply the main chamber. On day 2, 1 × 10^5^ endothelial cells were seeded in one of the lateral channels to mimic an endothelial vasculature. The channel was previously coated with collagen (1:20 in PBS buffer) for 40 min at 37 °C 5% pCO_2_. The endothelial cells were injected into the blood channel, and the chip was left in a vertical position to allow the cell attachment. Finally, the mixture medium was supplied on both channels, and the chip was maintained for 5 days after the endothelial seeding at 37 °C and 5% pCO_2_ before any experiment was performed. The cell culture medium was changed daily.

### 4.13. AuNPs@POM@PEG Permeability Assays in BBB-oC

To perform this assay, AuNPs@POM@PEG and AuNPs@PEG were previously labelled fluorescently with Alexa647. Before the injection of the nanoparticles, time 0 for the BBB-oC with the cell medium into channels was set, with the corresponding fluorescent settings. Each nanoparticle sample was then resuspended in cell media with a final 2.5 nM concentration (to achieve a good fluorescent signal) and added into the blood channel via independent devices. Then, the BBB-oC was placed into the fluorescent microscope (Nikon Ti2) and fluorescent images were captured until the end point of 1h. The recorded pictures were analyzed using Fiji/ImageJ^®^ software, and the permeability coefficients (cm/s) were determined following the equation from Campisi et al. [77].

For TEM visualization, AuNPs@POM@PEG were administered in the blood channel and incubated for 24 h at 37 °C and 5% CO_2_. Next, the channels were washed twice with PBS 1× to remove any nanoparticles that did not cross the BBB. The hydrogel was dissolved adding TRIzol reagent in both channels and incubated at 60 °C for 15 min. After that, the liquid in the main chamber could be collected. Holey carbon grids were prepared via depositing a droplet of the dissolved hydrogel onto them and letting them dry. Finally, the grids were observed with an FEI Magellan 400L XHR SEM, and EDX was performed on the region of interest.

## Data Availability

Data available on request. The data presented in this study are available on request from the corresponding author.

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
