# Peer review of "Polyoxometalate-Decorated Gold Nanoparticles Inhibit β-Amyloid Aggregation and Cross the Blood–Brain Barrier in a µphysiological Model"

_nanomaterials, 2023, doi:10.3390/nano13192697_

Round 1

Reviewer 1 Report

In the research article, titled ‘Polyoxometalate decorated gold nanoparticles inhibit β -amy-2 loid aggregation and cross the blood brain barrier in a μphysio-3 logical model.’, the authors combined gold nanoparticles (AuNPs) with polyethylene glycol (PEGs) and polyoxometalate (POM). They fully characterized the tri-component nanohybrid system and assessed its bioactivity through in vitro experiments, including assessments of biocompatibility (cell toxicity), the capacity of beta-amyloid (Aβ) aggregation inhibition, and the ability to cross the blood-brain barrier (BBB). The authors utilized a cell-cultured fluidic chip that mimicked the BBB, which is an innovative method worth further exploration. The experiments were designed and executed scientifically, and the data were presented appropriately. The English writing is fluent and easy to read.

However, compared to AuNPs@POMD-pep designed in previous research (PMID: 25376633), AuNPs@PEGs@POM appears to be a simpler imitation. Although the BBB-on-a-chip is relatively new and fits the theme of this special issue, it was first utilized by the authors in another research published this year (PMID: 36978078). Both of these limit the scientific value of this article.

There are several major concerns that the authors should consider:

1.      ‘There is only one example in the literature where AuNPs have been used in combination with POMs for the treatment of AD. However, in this case, POMs were used merely as linkers between Au nanorods (AuNRs) and the Aβ-targeted peptide inhibitor Aβ15-20’ This statement is not correct. N. Guo et al. had constructed AuNPs@POMD-pep (AuNPs: gold nanoparticles, POMD: polyoxometalate with WellsDawson structure, pep: peptide) as a multifunctional Aβ inhibitor, and the POMD (which is a specific type of POM) was utilized as an Aβ-binding inhibitor (PMID: 25376633). The POMD utilized in that article and the β1 isoform of POM used in this article were both validated in Aβ aggregation inhibition assays conducted by N. Guo et al. (PMID: 21433228). Since the two articles were cited, the authors should clarify the contributions of the two more clearly. Additionally, reference PMIDs: 30969760 and 24595206 should be added since they explored more aspects of the POM as an Aβ inhibitor.

2.      BBB-on-a-chip with Integrated micro-TEER for permeability evaluation of multi-functionalized gold nanorods against Alzheimer’s disease” has been formally published in the Journal of Nanobiotechnology in 2023 (PMID: 36978078), which describes the BBB-on-a-chip technology developed by the authors. This technology was utilized as a highlight in this article, therefore, it would be preferable to mention this research clearly in the Introduction and update the citation to its final version.

3.      There is a paucity of in vivo experiments, which calls for careful consideration and discussion of its limitations. The ability of the nanoparticles to cross the BBB-on-a-chip is not equivalent to much distribution to the brain when administered in vivo. Furthermore, the therapeutic effects of the nanoparticles have not been assessed, so the efficiency of AuNPs@PEGs@POM remains uncertain.

Minor comments:

1.      When reading this paper, some grammar mistakes, such as missing commas or using incorrect words, were found. These should be corrected for clarity:

â‘     In the title, “blood brain barrier” is better to be amended to “blood-brain barrier”.

â‘¡    In the Abstract, “nanotherapeutic system” is better to be amended to “nanotherapeutics”.

â‘¢    There is a missing comma before the conjunction “and” in lines 42, 200, and 260.

â‘£    In the introduction, some words or their forms should be corrected: “showed” in line 48, “antibody” in line 52, “beneficious” in line 56, “reduce proteins absorption” in line 62, “this” in line 87.

⑤    In line 220, “is separated of” should be corrected to “is separated from”.

â‘¥    In line 222, “is not harmful for” should be corrected to “is not harmful to”.

⑦    In line 224, “ranging 0.5 to 5 nM” should be corrected to “ranging from 0.5 to 5 nM”.

â‘§    In line 253, “inject in” is better to be amended to “inject into”.

⑨    All instances of “in vitro” should be written in italic form.

2.      The title of section 2.3 needs modification to include the part about cytotoxicity assays.

No.

Reviewer 2 Report

Generally, the proposed study is well-organized and presents interesting results.

The topic is actual and important, but some details need clarification/improvement/correction.

General comments:

Considering the FAIR principles (https://www.go-fair.org/fair-principles/) and recommendation for nanoscience (https://doi.org/10.1038/s41565-021-00911-6) it will we very beneficial for society and also authors sharing collected data as supplementary materials in some form of data file (like .xls)

Line 52-54

LEQEMBI is already approved by the FDA for the treatment of Alzheimer’s disease.

Line 78-80

I completely agree with the statement “a lack of prediction due to the physiological differences between humans and animals”. However, this topic should be described in detail. Especially in the context of AD when several animal models (especially small animals) do not suffer from this disease, and the artificially introduced changes are far from human AD pathophysiology.

Line 166-168

It is not clear why Dh I pZ are not given to AuNPs@POM .

Line 250

The general overview of transport of AuNPs thought BBB should be added. Is the transport of AuNPs by BBB driven by size or many other NPs features/properties? Is there some critical value of size or zeta potential for transport by BBB?
